Impacts of the invasive shot hole borer (Euwallacea kuroshio) are linked to sewage pollution in southern California: the Enriched Tree Hypothesis

Boland John M. 1 JohnBoland@sbcglobal.net
Woodward Deborah L. 2
1 Boland Ecological Services , San Diego, CA , USA
2 California Water Quality Control Board, San Diego Region , San Diego, CA , USA
Anderson Todd
Electronic publication date: 2019 May 1
Publication date: 2019
Volume: 7
Electronic Location ID: e6812
Received 2018 Nov 21; Accepted 2019 Mar 19
Copyright: © 2019 Boland and Woodward
Copyright year: 2019
Copyright holder: Boland and Woodward
License: This is an open access article distributed under the terms of the Creative Commons Attribution License, which permits unrestricted use, distribution, reproduction and adaptation in any medium and for any purpose provided that it is properly attributed. For attribution, the original author(s), title, publication source (PeerJ) and either DOI or URL of the article must be cited.
License URL: https://creativecommons.org/licenses/by/4.0/

Keywords: Arroyo willow, Black willow, Kuroshio Shot Hole Borer, Salix gooddingii, Salix lasiolepis, Sewage spills, Wood density, Wood moisture content

Funding: Department of Navy on behalf of the Naval Base Coronado via a Cooperative Agreement between the US Navy and the US Fish and Wildlife Service and the Southwest Wetlands Interpretive Association F16AC01065 John Boland received funding from the Department of Navy on behalf of the Naval Base Coronado via a Cooperative Agreement (F16AC01065) between the US Navy and the US Fish and Wildlife Service and the Southwest Wetlands Interpretive Association. Deborah Woodward received no funding for this work. The funders had no role in study design, data collection and analysis, decision to publish, or preparation of the manuscript.

==============================
The Kuroshio Shot Hole Borer (KSHB, Euwallacea kuroshio) and the Polyphagous Shot Hole Borer (E. whitfordiodendrus; Coleoptera: Curculionidae: Scolytinae) have recently invaded southern California and are attacking live trees in commercial agriculture groves, urban parks and native riparian forests. Among native forests the worst impacts observed to date have been in the Tijuana River Valley in south San Diego County, where approximately 30% of the native willows (Salix spp.), or 120,000 trees, have died as a result of a KSHB infestation. This paper examines wood densities, wood moisture contents, KSHB infestation rates, and KSHB-induced mortality rates in two willow species (Salix lasiolepis and S. gooddingii) at sites near and far from sewage input. Comparisons were made on two spatial scales: broadly among sites within San Diego County; and locally among sites within the Tijuana River Valley. The results showed that, on average, willow trees growing closest to sewage pollution had significantly lower wood density, higher wood moisture content, higher KSHB infestation rates, and higher KSHB-induced willow mortality rates than those growing farther away. We present the Enriched Tree Hypothesis to explain the link between sewage pollution and KSHB impacts; it is as follows: (A) Riparian trees subject to nutrient enrichment from frequent sewage pollution grow quickly, and their fast growth results in wood of low density and high moisture content. If attacked by the KSHB, the trunks and branches of these nutrient-enriched trees provide an environment conducive to the fast growth of the symbiotic fungi upon which the KSHB feeds. With an abundant food supply, the KSHB population increases rapidly and the trees are heavily damaged by thousands of KSHB galleries in their trunks and branches. (B) Riparian trees not subject to frequent sewage pollution grow more slowly and have denser, drier wood. Conditions in their trunks and branches are not conducive to the fast growth of the KSHB’s symbiotic fungi. The KSHB generally ignores, or has low abundances in, these slow-growing trees. This new hypothesis explains current patterns of KSHB impact in San Diego County and focuses attention on the important roles of the environment and preexisting conditions of trees in determining the extent of KSHB impact. It highlights the Tijuana River Valley as an unusual site due to high sewage inputs and predicts that the high KSHB-induced willow mortality seen there should not occur in other natural riparian habitats in southern California. Most importantly, by identifying sewage pollution (or nutrient enrichment) as a major risk factor for KSHB impacts, the hypothesis ratchets down the KSHB-threat level for most riparian sites in southern California and directs attention to other nutrient-enriched sites as those most at risk.

Introduction

Two closely related ambrosia beetle species have recently invaded southern California and are attacking live trees in commercial agriculture groves, urban parks, and native forests (Eskalen et al., 2013; Freeman et al., 2013; Boland, 2016; Eskalen, 2018). They are the newly described Kuroshio Shot Hole Borer (KSHB, Euwallacea kuroshio Gomez and Hulcr, sp. nov.) and Polyphagous Shot Hole Borer (PSHB, Euwallacea whitfordiodendrus Schedl 1942, stat. rev.; Coleoptera: Curculionidae: Scolytinae; Stouthamer et al., 2017; Gomez et al., 2018). These borers are considered a serious threat because of the many important tree species they can use as reproductive hosts (Eskalen, 2018) and because of the considerable impacts to the Tijuana River Valley forests, where an estimated 120,000 willows (Salix spp.), or approximately 30% of the native trees, died in just three years due to a KSHB infestation (Boland, 2016, 2018).

The PSHB and KSHB were first observed in 2003 and 2013, respectively, and have since been caught in traps or on trees in coastal southern California and Baja California, Mexico (Eskalen, 2018). They are not advancing as a diffusion-like expanding front (sensu Wilson et al., 2009), but instead are appearing at disjunct sites in the region (Eskalen, 2018). When present they infest and impact sites to differing degrees and, because so little is known about the ecology of the borers, it has been impossible to explain their current patterns of distribution and impact, or to predict where they will impact next.

The Tijuana River Valley is unusual in two respects—it is the site where the KSHB has had its greatest impact and it appears to be the most polluted valley in southern California. As the pollution is sewage, a plant fertilizer, we suggest that there is a link between the nutrient enrichment of willows and their susceptibility to shot hole borer attack. This is not a new idea; Gadd (1944a, 1944b) described a controlled experiment in which different fertilizers were added to tea bushes (Camellia sinensis L.) in Ceylon and reported that “treatments, which improve the yielding capacity of the tea bush, also increase the liability of the bush to attack by the shot hole borer beetle” (Gadd, 1944b: 253). The beetle was the Tea Shot Hole Borer (Euwallacea fornicatus Eichhoff), and Gadd went on to state, “Why this should be so raises a question of considerable importance, to which no very exact answer can be given at present.” Little has changed since then. The link between nutrient enrichment of the soil and the susceptibility of trees to shot hole borer attack has not been made clear and has led Hulcr & Stelinski (2017: 296), in their recent review, to comment that “in ambrosia beetle research, the role of the environment and preexisting conditions of the trees has not yet been well appreciated, even though it appears to determine the impact of these beetles.”

In this study, we examined the links between the environment (i.e., sewage loading), preexisting conditions (i.e., the characteristics of willow wood), and the severity of the KSHB impacts in two species of willows growing in the Tijuana River Valley and in other sites in San Diego County. In particular, we evaluated the following hypotheses: (H1) the level of sewage pollution in the Tijuana River Valley is greater than at other sites in San Diego County; (H2) willow trees growing in the Tijuana River Valley have wood characteristics that are significantly different to those growing outside the valley; (H3) willow trees growing in the Tijuana River Valley have KSHB infestation rates and KSHB-induced mortality rates that are significantly greater than those growing outside the valley; (H4) a site with intermediate nutrient enrichment will have intermediate wood characteristics and intermediate shot hole borer impacts; and (H5) the links between sewage pollution level, willow wood characteristics, and KSHB impacts observed on a large scale (on the order of km) will also hold on a small scale (on the order of m).

Based on the results of these tests, we have developed a general hypothesis describing the link between nutrient enrichment and shot hole borer impacts. This hypothesis explains the current patterns of KSHB distribution and impact, and allows one to make predictions about where the KSHB will cause impacts next. While the tests focused on the KSHB in San Diego County, the insights gained are likely to apply to both the KSHB and PSHB throughout southern California.

Materials and Methods

The shot hole borers

The KSHB and PSHB weaken or kill trees through their tunneling activities and their associated fungal symbionts (Eskalen et al., 2013; Freeman et al., 2013; Eskalen, 2018). Females drill into tree trunks and branches, excavate galleries in the wood, inoculate their gallery walls with fungi (e.g., Fusarium sp.), and live in their galleries eating the fungi and reproducing (Biedermann, Klepzig & Taborsky, 2009). Within a few weeks mated females emerge and either remain on their natal tree or fly to new trees perpetuating the infestation (Rudinsky, 1962). The borers are small (∼2 mm in length), but their many tunnels and spreading fungal symbionts can undermine the structural integrity of the tree (Boland, 2016) or block the water-transporting vessels in the xylem and cause the progressive death of shoots and branches (Eskalen et al., 2013). Both borers are currently rated “Q” by the California Department of Food and Agriculture, indicating that more information is needed before they can be correctly rated. Most research on KSHB and PSHB to date has focused on controlling the borers, especially in agricultural and urban settings (Dodge et al., 2017; Mayorquin et al., 2018) and on describing their fungal symbionts (Freeman et al., 2013; Na et al., 2018). The borers causing the damage in the riparian habitats of the Tijuana River Valley were collected and identified as the KSHB by Dr. Akif Eskalen at University of California Riverside (UCR), and specimens have been stored in the UCR collection.

The willows

The arroyo willow (Salix lasiolepis Benth.) and black willow (Salix gooddingii C.R. Ball) are abundant riparian trees in San Diego County. They grow to heights of 9 and 20 m, respectively, and dominate early successional riparian forests along the coastal rivers and streams. Recent research on these willows has addressed seed dispersal (Boland, 2014a), seedling establishment patterns (Boland, 2014b, 2017a), adult tree zonation (Boland, 2014b), KSHB impacts (Boland, 2016), and post-KSHB recovery (2018), and all has been conducted in the Tijuana River Valley.

The main study site—the Tijuana River Valley

The Tijuana River Valley is a coastal floodplain in San Diego County, California, of approximately 15 square kilometers at the end of a 4,480 square kilometer watershed (Fig. 1). The Tijuana River is an intermittent stream (Boland, 2014b) that frequently becomes polluted with sewage and other wastewater as it flows through the City of Tijuana, Mexico, before entering the valley (TRNERR (Tijuana River National Estuarine Research Reserve), 2010). Its channel through the valley is not armored and, because it has frequently changed course during the past 40 years (Safran et al., 2017), the riparian forests in the valley are a mosaic of stands of different ages at various distances from the current river flows (Boland, 2016). The riparian forests are dominated by arroyo and black willows (Boland, 2014b) and are preserved within three adjoining open space parks. The riparian habitats support many species, most notably the endangered least Bell’s vireo (Vireo bellii pusillus) for which most of the riparian habitat is designated critical habitat (U.S. Fish and Wildlife Service, 1994). For access to their properties in the Tijuana River Valley and elsewhere, we obtained permission from the US Fish and Wildlife Service (permits 19001TJS, 2017-81720-R-011), State of California Parks and Recreation (17-669-01, 17-669-02), County of San Diego Department of Parks and Recreation (ROE08.08.16-08.31.19, TRVRP07-2017, MP09-2017), City of San Diego (07.23.2018SS), and Orange County Parks (08.04.2017CN).

Figure 1 Site locations.

Locations of the 18 sampling sites in San Diego County, California. Sites influenced by polluted Tijuana River flows are in blue, and those not influenced by polluted Tijuana River flows are in yellow. Nomenclature follows Table 1. This map was created using ArcGIS® software; image source: Esri, DigitalGlobe, GeoEye, i-cubed, USDA FSA, USGS, AEX, Getmapping, Aerogrid, IGN, IGP, swisstopo, and the GIS User Community.

The level of sewage pollution in the Tijuana River Valley is greater than at other sites in San Diego County (H1)

Sewage spill data were compiled for the Tijuana River Valley and the rest of San Diego County for the 3-year period January 1, 2015–December 31, 2017. Spill data for the Tijuana River Valley were from the International Boundary and Water Commission’s spill reports, which document “dry weather transboundary wastewater” flows from Mexico that contain raw sewage, partially treated sewage and other wastewaters (San Diego Regional Water Quality Control Board, 2018). Spill data for the rest of San Diego County were from the California State Water Resources Control Board sanitary sewer overflow incident web site (State Water Board, 2018). Each incident report was examined, and the spill volume minus the recovered volume was recorded for all spill volumes >10,000 gallons that occurred in the San Diego County area south of Oceanside, an area of approximately 3,000 square kilometers.

To confirm that sewage spills were influencing the nutrient concentrations in the rivers, water quality data from the California Environmental Data Exchange Network (CEDEN) website were examined from the one station in the Tijuana River Valley (Tijuana River 5) and from six stations in San Diego County (TWAS 1 on the Otay River, TWAS 1 on the Sweetwater River, TWAS 1 and 2 on the San Diego River, TWAS 2 and 6 on Peñasquitos Creek; (CEDEN (The California Environmental Data Exchange Network), 2018)). The maximum values for “Ammonia as N, total” and “Phosphorus as P, total” from January 2005 to December 2017 were recorded for each station.

Willow trees growing in the Tijuana River Valley have wood characteristics that are significantly different to those growing outside the valley (H2)

Nine sites were chosen along the main river channel in the Tijuana River Valley and nine sites were chosen away from the influence of the Tijuana River. When the Tijuana River Valley was first impacted by the KSHB in 2015 the riparian habitats were divided for study purposes into 29 vegetation units so that each unit was relatively homogenous in terms of plant density, species composition and age (Boland, 2016). Nine of the forest units were chosen for this study in a stratified random manner and their unit numbers were retained (Fig. 1; Table 1). The other nine sites, chosen to represent the rest of San Diego County, were mainly in other watersheds, though one site (OUT-1) was within the Tijuana River Valley but not influenced by the main river flows. All of the San Diego County sites were in open space parks with established riparian forests comparable to those in the Tijuana River Valley.

Table 1 Site descriptions.

The park in which each site is located and its associated river. For sites within the Tijuana River Valley, the distance to the Tijuana River (TJR) main channel is given.

Site	Park	River	Distance to TJR main channel (m)	
2	Tijuana River Valley Regional Park	Tijuana River	0	
3	Tijuana River Valley Regional Park	Tijuana River	0	
9	Tijuana River Valley Regional Park	Tijuana River	0	
12	Tijuana River Valley Regional Park	Tijuana River	0	
13	Tijuana River Valley Regional Park	Tijuana River	40	
15	Tijuana River Valley Regional Park	Tijuana River	100	
17	Tijuana River Valley Regional Park	Tijuana River	140	
19	Tijuana River Valley Regional Park	Tijuana River	60	
21	Tijuana Slough National Wildlife Refuge	Tijuana River	520	
OUT-1	Border Field State Park	Tijuana River tributary	1,400	
OUT-2	Otay Valley Regional Park	Otay River	Outside TJR Valley	
OUT-3	Pacific Gateway Park	Tijuana River tributary	Outside TJR Valley	
OUT-4	Balboa Park	Florida Canyon Creek	Outside TJR Valley	
OUT-5	San Diego National Wildlife Refuge	Sweetwater River	Outside TJR Valley	
OUT-6	Mission Valley Preserve	San Diego River	Outside TJR Valley	
OUT-7	Mission Trails Regional Park	San Diego River	Outside TJR Valley	
OUT-8	Torrey Pines State Natural Reserve	Peñasquitos Creek	Outside TJR Valley	
OUT-9	Los Peñasquitos Canyon Preserve	Peñasquitos Creek	Outside TJR Valley	

Wood density was determined from branch samples of live arroyo and black willow trees collected between August 28 and October 10, 2017 at the 18 sites; all trees still had their leaves and were actively growing. Branch samples were 2.3–3.0 cm in diameter, approximately 23 cm long, not damaged or infested, and estimated to be 3–5 years old. Branch samples were cut with a small pruning saw on a walk through the forest, one sample per tree, usually until ten arroyo and ten black willows were collected. Cut ends on the trees were immediately sprayed with TreeKote Tree Wound Dressing to reduce the risk of infection at the cut, and the saw was cleaned between samples with a disinfectant wet wipe. There was some variation in the number of samples collected (Table 2): at Sites 2 and 19 more than ten samples were collected; at Site 3 arroyo willows were absent so none could be collected; and at Site OUT-8 black willows were absent. Branch samples were taken to the lab, stripped of their bark, and measured for wet weight and volume. Volume was measured using the water displacement method (Chave, 2005). The dry weight of each sample was recorded after 3 days in a drying oven at 102–104 °C, and wood density was calculated as: Density (g/ml) = Dry weight (g)/Volume (ml). A total of 372 branch samples were collected and analyzed for wood density (Table 2).

Table 2 Number of samples.

The number of branch samples analyzed for wood density and wood moisture content. SALA = Salix lasiolepis (arroyo willow), and SAGO = Salix gooddingii (black willow).

Site	Wood density	Wood moisture content	
	SALA	SAGO	SALA	SAGO	
2	20	22	20	10	
3	0	10	0	10	
9	10	10	10	10	
12	10	10	7	7	
13	10	10	7	7	
15	10	10	10	7	
17	10	10	10	10	
19	20	10	0	0	
21	10	10	7	7	
OUT-1	10	10	10	10	
OUT-2	10	10	10	10	
OUT-3	10	10	10	5	
OUT-4	10	10	10	10	
OUT-5	10	10	10	8	
OUT-6	10	10	10	10	
OUT-7	10	10	10	10	
OUT-8	10	0	10	0	
OUT-9	10	10	10	10	
Total	190	182	161	141	
Grand Total	372	302	

The moisture content of each branch sample was calculated as: Moisture content (%) = (Wet weight (g) − Dry weight (g)) × 100/Wet weight (g). This is the percentage of the original green weight that was water and is described as “moisture content on a wet basis” (Hartley & Marchant, 1995). In total, 302 branch samples were analyzed for moisture content. (Wet weight was not recorded for the first round of samples collected, resulting in fewer samples analyzed for moisture content than for wood density (Table 2)).

Wood densities and moisture contents of the willows in the Tijuana River Valley were compared to those in the San Diego County sites, and tested separately for each willow species via a two-level nested ANOVA for unequal sample sizes (McDonald, 2009 and the worksheets therein). The null hypothesis was that there were no differences in the wood characteristics for the two groups of sites.

Willow trees growing in the Tijuana River Valley have KSHB infestation rates and KSHB-induced mortality rates that are significantly greater than those growing outside the valley (H3)

The impact of the KSHB was measured at the nine sites along the main Tijuana River channel and the nine San Diego County sites. At each site the condition of as many willows as could be examined in two hours was recorded. For infestation rate, each willow was examined and classified as “currently infested” if it had evidence of active tunneling by the KSHB, such as extrusion of sawdust or recent gumming out of sap from KSHB holes. For mortality rate, each willow was classified as alive, recently dead from KSHB attack if it had KSHB holes, or dead from some other cause if it had no KSHB holes. KSHB infestation rates and KSHB-induced mortality rates were calculated for each site as the percent of the total number of willow trees examined. At 13 sites, these surveys were done during October 2017. At the other five sites (Tijuana River Valley sites 2, 3, 9, 12, and 13), such surveys could not be done in 2017 because of extensive willow damage caused by the KSHB in 2015–2016 (Boland, 2016); instead, for these sites, infestation rates are from Boland (2016) and mortality rates are from Boland (2017b). KSHB infestation rates and KSHB-induced mortality rates within the two groups of sites were compared using the Mann–Whitney U-test (McDonald, 2009).

A site with intermediate nutrient enrichment will have intermediate wood characteristics and intermediate shot hole borer impacts (H4)

To include a nutrient-enriched site outside the Tijuana River Valley, we surveyed two sites in Orange County. These were Fairview Park, a constructed treatment wetland consisting of riparian trees, and South Talbert Regional Park, a natural riparian habitat downstream of Fairview Park. As the water flowing into these sites is nutrient-enriched (Brown, 2018), these sites should have wood densities and shot hole borer impacts that are intermediate between the heavily polluted Tijuana River and the less polluted other San Diego County sites. We collected 20 branch samples from arroyo willows at each of these sites in August 2017 and processed the samples as described above, except we determined only wood density, and not moisture content. During collection we noted whether the sampled tree was infested with shot hole borer (PSHB in these sites; Eskalen, 2018).

The links between sewage pollution level, willow wood characteristics, and KSHB impacts observed on a large scale (on the order of km) will also hold on a small scale (on the order of m) (H5)

The 10 sites within the Tijuana River Valley (Sites 2 to 21 and OUT-1; Table 1) were compared with respect to the level of sewage pollution, wood density, wood moisture content, KSHB infestation rate, and KSHB-induced mortality rate using correlation analysis (McDonald, 2009). For this small-scale comparison, data for arroyo and black willows were combined. Distance from the main Tijuana River channel was used as a proxy for the amount of sewage reaching each site, the assumption being that the greater the distance from the main channel, the lower the exposure to sewage-polluted flows (Table 1). To present logarithmic trend lines, one meter was added to all distances from the main channel to remove zero distances.

Results

The level of sewage pollution in the Tijuana River Valley is higher than at other sites in San Diego County (H1)

The total sewage spill volume into the Tijuana River Valley was approximately 30 times that of the entire rest of San Diego County during the 3-year period 2015–2017. In that period, there were 55 cross border flows from Mexico into the Tijuana River totaling more than 220 million gallons (Fig. 2A) and only seven sewage spills totaling less than eight million gallons within the rest of San Diego County (Fig. 2B). The largest cross border flows into the Tijuana River Valley were of 143 and 27 million gallons (off the scale on Fig. 2A), whereas the largest spill in San Diego County outside the Tijuana River Valley was of seven million gallons into a concrete lined channel that discharged into Mission Bay (Fig. 2B). CEDEN water quality measurements reflected the sewage spill data; the Tijuana River values for maximum nitrogen and phosphorus were 23.40 and 9.75 mg/l, respectively, and were 10–20 times more than the maximum measured in the other San Diego County rivers (1.20 and 1.01 mg/l). The hypothesis was supported: the level of sewage pollution, or nutrient enrichment, was far greater in the Tijuana River Valley than elsewhere in the county.

Figure 2 Sewage spill volumes.

Magnitude of sewage input during the three year period 2015–2017 into two areas. (A) The Tijuana River. (B) The rest of San Diego County.

Willow trees growing in the Tijuana River Valley have wood characteristics that are significantly different to those growing outside the valley (H2)

Wood density of branch samples ranged in black willows from 0.29 to 0.53 g/ml (n = 182) and in arroyo willows from 0.31 to 0.56 g/ml (n = 190). For both willow species, wood density was significantly lower at the polluted Tijuana River sites than at the less polluted San Diego County sites (Fig. 3). For black willows, the mean (±S.D.) wood densities at the polluted and less polluted sites were 0.40 (±0.043) g/ml (n = 102) and 0.44 (±0.038) g/ml (n = 80), respectively, and these densities were significantly different (ANOVA F = 16.8; p < 0.001). For arroyo willows, the mean wood densities at the polluted and less polluted sites were 0.42 (±0.043) g/ml (n = 100) and 0.48 (±0.042) g/ml (n = 90), respectively, and these densities were significantly different (ANOVA F = 21.5; p < 0.001). Wood densities of black willows and arroyo willows were significantly correlated at the sites where both species were present (r = 0.886, n = 15 sites, p < 0.0001), that is, where wood density was low in one species it was low in the other, and where it was high in one species it was high in the other.

Figure 3 Willow wood densities.

Densities (g/ml) of branch samples from the two willow species collected at the polluted Tijuana River (left) and less polluted San Diego County (right) sites. (A) Black willow. (B) Arroyo willow. Data are mean ± 1 std. dev. Site nomenclature follows Table 1.

Wood moisture content of branch samples ranged in black willows from 42% to 67% (n = 141) and arroyo willows from 44% to 63% (n = 161). For both willow species, wood moisture contents from the polluted Tijuana River sites were significantly higher than those from the less polluted San Diego County sites (Fig. 4). For black willows, the mean wood moisture contents at the polluted and less polluted sites were 54% (±4.7%, n = 68) and 49% (±3.4%, n = 73), respectively, and these moisture contents were significantly different (ANOVA F = 17.1, p < 0.005). For arroyo willows, the mean moisture contents at the polluted and less polluted sites were 55% (±3.7%, n = 71) and 50% (±3.2%, n = 90), respectively, and these moisture contents were significantly different (ANOVA F = 35.8, p < 0.001). Like wood density, moisture content in black willows and arroyo willows were significantly correlated at the sites where both species were present (r = 0.682, n = 15 sites, p < 0.01), that is, where moisture content was high in one species it was high in the other, and where it was low in one species it was low in the other.

Figure 4 Willow wood moisture contents.

Moisture contents (%) of branch samples from the two willow species collected at the polluted Tijuana River (left) and less polluted San Diego County (right) sites. (A) Black willow. (B) Arroyo willow. Data are mean ± 1 std. dev. Site nomenclature follows Table 1.

The hypothesis was supported: the wood characteristics (wood densities and moisture contents) of black and arroyo willows growing in the polluted Tijuana River Valley sites were significantly different to those of willows growing in the less polluted sites.

Willow trees growing in the Tijuana River Valley have KSHB infestation rates and KSHB-induced mortality rates that are significantly greater than those growing outside the valley (H3)

KSHB infestation rates and KSHB-induced willow mortality rates ranged from 0% to 100% and 0% to 97%, respectively, at the 18 sites (Table 3). Willows in the polluted Tijuana River sites had significantly higher rates of KSHB infestation (Mann–Whitney U-test; n = 9, 9; U = 0; p < 0.001) and significantly higher rates of KSHB-induced mortality (Mann–Whitney U-test; n = 9, 9; U = 4.5; p < 0.01). The hypothesis was therefore supported.

Table 3 Impacts of the KSHB.

KSHB infestation rates and KSHB-induced willow mortality rates at sites where branch samples were collected.

Site	# Willows examined	% Infested	% Mortality	
A. Polluted sites	
2	187	94**	67*	
3	65	100**	97*	
9	57	100**	42*	
12	36	100**	78*	
13	49	97**	41*	
15	69	17	0	
17	37	93	3	
19	34	83	12	
21	79	86	4	
Median	53	84	41	
B. Less polluted sites	
OUT-1	48	0	0	
OUT-2	63	0	0	
OUT-3	79	0	0	
OUT-4	82	9	0	
OUT-5	58	0	0	
OUT-6	68	0	0	
OUT-7	42	0	0	
OUT-8	42	0	0	
OUT-9	72	0	0	
Median	63	0	0	
Notes:

* Boland (2017b).

** Boland (2016).

Among all sites, willow mortality rates were significantly and negatively correlated with average wood densities (r = 0.903, n = 18, p < 0.0001), and significantly and positively correlated with average wood moisture content (r = 0.908, n = 17, p < 0.0001). That is, KSHB caused the most damage at sites with trees of low wood density and high moisture content.

A site with intermediate nutrient enrichment will have intermediate wood characteristics and intermediate shot hole borer impacts (H4)

Wood densities of the arroyo willow branch samples ranged from 0.33 to 0.53 g/ml (n = 40) at the two Orange County sites that receive polluted runoff. For Fairview Park, the mean wood densities were 0.43 (± 0.050) g/ml (n = 20) and for South Talbert they were 0.44 (± 0.039) g/ml (n = 20). The wood characteristics of the arroyo willow were therefore intermediate between the polluted Tijuana River sites and the less polluted San Diego County sites (Fig. 5). PSHB infestation rates at the two Orange County sites were also intermediate; the arroyo willows at Fairview Park were 70% infested (n = 20) and the arroyo willows at South Talbert were 55% infested (n = 20). The Orange County results are therefore consistent with the hypothesis.

Figure 5 Inclusion of two intermediates.

The density (g/ml) of arroyo willow branch samples collected at two Orange County sites (center) compared to densities collected at the polluted Tijuana River sites (left) and the less polluted San Diego County sites (right). FV, Fairview Park; ST, South Talbert Regional Park. Data are mean ± 1 std. dev.

The links between sewage pollution level, willow wood characteristics, and KSHB impacts observed on a large scale (on the order of km) will also hold on a small scale (on the order of m) (H5)

Among the ten sites within the Tijuana River Valley there were significant patterns in wood densities, wood moisture contents, KSHB infestation rates, and willow mortality rates. Wood densities of willow branch samples were significantly lower near the main channel than at more distant sites (r = 0.278, n = 222 trees, p < 0.0001). Moisture contents of branch samples were significantly higher near the main channel (r = 0.316, n = 159 trees, p < 0.0001). KSHB infestation rates were significantly higher near the main channel (r = 0.723, n = 10 sites, p < 0.05), and willow mortality rates due to the KSHB were high near the main channel and dropped precipitously to zero away from the channel (the values for the logarithmic trend line were: r = 0.888, n = 10 sites, p < 0.01). On the other hand, there were no significant patterns in stand structural traits with distance from the main channel; stand age (r = 0.370, n = 10 sites, p > 0.05), median girth of willows (r = 0.191, n1 = 21–40 trees per site, n2 = 10 sites, p > 0.05), and stand density of willows (r = 0.191, n = 10 sites, p > 0.05) showed no significant patterns among the 10 sites.

The hypothesis was supported: the links between pollution level, willow wood characteristics, and KSHB impacts observed on a large scale (within San Diego County) also hold on a small scale (within the Tijuana River Valley), that is, willows growing in or close to the sewage-polluted water had wood of lower density and higher moisture content, and suffered higher rates of KSHB infestation and KSHB-induced mortality than those growing farther away.

Discussion

In this study, high levels of sewage pollution were correlated with high rates of KSHB infestation and willow mortality, as well as with willow wood of low density and high moisture content. These correlations were observed both countywide and locally. We propose the following new hypothesis, the Enriched Tree Hypothesis, to explain this connection between sewage pollution and the KSHB: Riparian trees subject to nutrient enrichment from frequent sewage pollution grow quickly, and their fast growth results in wood of low density and high moisture content. If attacked by the KSHB, the trunks and branches of these nutrient-enriched trees provide an environment conducive to the fast growth of the symbiotic fungi upon which the KSHB feeds. With an abundant food supply, the KSHB population increases rapidly and the trees are heavily damaged by thousands of KSHB galleries in their trunks and branches.

Riparian trees not subject to frequent sewage pollution grow more slowly and have denser, drier wood. Conditions in their trunks and branches are not conducive to the fast growth of the KSHB’s symbiotic fungi. The KSHB generally ignores, or has low abundances in, these slow-growing trees.

In short, trees of the same species growing under different conditions have different wood characteristics, and the most enriched trees among them are the most susceptible to an abundant KSHB infestation.

Each element of the Enriched Tree Hypothesis has support in the literature. Sewage contains the most important plant macronutrients—nitrogen, phosphorus, and potassium (Singh & Agrawal, 2008)—and, when used as a fertilizer, sewage is as effective as chemical fertilizers in promoting tree growth (Prescott & Blevins, 2005). Nutrient enrichment is known to influence wood characteristics, and heavy fertilization and irrigation leads to fast growth, low density wood, and high moisture content in several species of trees (Constantz & Murphy, 1990; Hacke et al., 2010; Drew, Downes & Evans, 2011; West, 2014). With respect to ambrosia beetle growth, both the nutrient and moisture contents of wood are known to influence beetle growth. The standard medium for growing ambrosia beetles and their symbiotic fungi in the laboratory is water-based and rich in nutrients (Biedermann, Klepzig & Taborsky, 2009; Cooperband et al., 2016). The growth rates of the fungi and larvae of ambrosia beetles are positively correlated with the moisture content of the host plants (Rudinsky, 1962); and several authors, working on bark beetles, have shown that low nitrogen levels adversely affect beetle development by increasing development time and reducing fecundity (Kirkendall, 1983; Ayres et al., 2000; Raffa, Gregoire & Lindgren, 2015). Finally, when the KSHB find a suitable host they attack it en masse; one black willow, killed by the KSHB in the Tijuana River Valley, had up to 40 KSHB holes per 40 cm2 of bark and more than 20,000 KSHB holes over its entire surface (Boland, 2017b).

The results of the tests are also corroborated by the findings of others. Several researchers have found that the Tijuana River is the most polluted in San Diego County (Tetra Tech, Inc., 2006; McLaughlin et al., 2014) and that the KSHB’s impact has been greatest in the Tijuana River Valley—the U.S. Fish and Wildlife Service has conducted extensive surveys for KSHB throughout San Diego County and have reported that “nowhere has been as severely hit (by the KSHB) as the Tijuana River Valley” (Eric Porter, USFWS, personal communication March 6, 2018). Also, black willows growing in the Tijuana River Valley were found to grow 50% faster than black willows growing in the Otay River watershed (Boland, 2018). Finally, the measured range of wood density for black and arroyo willows in this study (0.29–0.56 g/ml) is comparable to the range in the literature for Salix spp. (0.4–0.6 g/ml; Engineering Tool Box, 2018) except, notably, willows growing in the nutrient enriched Tijuana River Valley extend the lower range by more than 0.1 g/ml.

The Enriched Tree Hypothesis can account for the disjunct distribution of shot hole borers in southern California, and it can explain the differing severity of shot hole borer impacts among infested sites. The hypothesis predicts that nutrient enriched sites will have more extensive infestations and impacts, and it predicts that the progression, or trajectory, of an infestation will vary among sites depending on the level of nutrient enrichment and tree condition. An alternative view, sometimes informally expressed at meetings, is that an infested site will always become increasingly infested and increasingly impacted over time and, hence, an infestation in a few trees in a stand is said to be a young infestation and an infestation in many trees is said to be an older infestation. This view assumes that (a) host trees of a given species are of similar condition in all stands, and (b) the trajectory of an infestation will be the same in all stands. Our results, especially those from within the Tijuana River Valley, do not support these assumptions because (a) tree condition in terms of wood density and moisture content varied significantly among willow stands in the valley, and (b) willows close to the sewage-polluted river were quickly and heavily infested and suffered high mortality, whereas willows farther from the river have remained only lightly infested and largely undamaged. While age of infestation must play some role, nutrient enrichment and tree condition appear to play the primary roles in determining the trajectory and ultimate severity of an infestation.

Understanding the mechanism underlying the rapid growth of ambrosia beetles in enriched trees could be a promising area of future research. Fungi require sugars and nutrients (such as nitrogen and phosphorus) for growth (Kendrick, 1992). In enriched trees it may be as simple as the sap in the tree is nutrient loaded in two ways—phloem sap is loaded with sugars from the fast-growing leaves, and xylem sap is loaded with nutrients from the enriched soil—and these extremely high nutrient conditions in the trunks and branches promote the fast growth of the symbiotic fungi and, ultimately, the fast growth of the beetles. Measurements of the sugar and nutrient concentrations and loading rates in trees subject to differing amounts of sewage, followed by replicated and controlled growth rate trials with the fungal symbionts receiving various appropriate sugar and nutrient concentrations and loading rates would be particularly valuable.

Predicting where KSHB will occur next

One rationale for this study was to answer the questions: “Where are the shot hole borers likely to appear next? And where are they going to do the most damage?” Until now, these questions have been addressed using the list of plant species that the KSHB and PSHB can use as reproductive hosts (Eskalen, 2018). The logic has been that if a tree species is on that list, then all individuals of that species, at all sites, are at risk of being killed by an infestation. McPherson et al. (2017) used this logic to predict that nearly 12 million urban trees were at risk in coastal southern California and the asset value lost due to the invasive shot hole borers could be greater than $9 billion over 10 years. By the current logic, many millions of riparian trees in southern California are at risk of being killed by the shot hole borers, and all of these trees have the same high risk.

Now, using the Enriched Tree Hypothesis in a predictive manner, risk assessments can be improved. By identifying sewage pollution (or nutrient enrichment) as a major risk factor for KSHB impacts, predictions should now be based on the presence of a susceptible plant species plus the presence of sewage pollution (or overfertilization). Only areas with both of these factors will be at high risk of infestation; all other areas will be at lower risk. This refinement immediately ratchets down the threat level for most riparian sites in San Diego County. A few natural sites may have some nutrient enrichment leading to moderately enriched trees and moderate infestations, but high nutrient conditions are rare in natural habitats and are more likely to be found elsewhere, in treatment wetlands, agricultural fields, golf courses, and urban parks. This refinement also strongly suggests that the Tijuana River Valley is an unusual case, and that the extreme sewage pollution in the Tijuana River makes the trees growing near the main channel unique in the region. The rapid invasion by the KSHB, the subsequent collapse of the willow canopy, and the many thousands of willow deaths in the valley (Boland, 2016, 2017b, 2018) should not be expected to occur in more normal, unpolluted riparian habitats elsewhere. The maximum level of impact to be expected in moderately enriched sites is likely to be similar to the Orange County sites, with their intermediate pollution levels and intermediate infestation rates.

The Enriched Tree Hypothesis should make one examine more closely the dire predictions others have made about the future impact of the borers in southern California. For instance, according to the Enriched Tree Hypothesis the predictions that the borers will (1) damage other natural sites as severely as the Tijuana River Valley (Greer, Rice & Lynch, 2018), (2) kill all the sycamores (Platanus racemosa Nutt.) in California (Raver, 2018), and (3) cause billions of dollars worth of damage to urban trees (McPherson et al., 2017) likely overstate the threat.

The conventional way to answer the question “Where are the borers most likely to appear next?” is to construct a pest risk map using host distributions, empirical measures of cold tolerance, and models of climate suitability (Venette et al., 2010). Maps of this sort would predict the final geographic range of the spreading insect and guide management efforts in those areas, but they have two drawbacks. First, they usually do not indicate where the impacts will be greatest within the range; as Venette, Coleman & Seybold (2015: 237) state, “the magnitude of impact remains the most difficult component of any pest risk assessment to forecast.” Second, these maps assume that all individuals of a host tree species are the same (McPherson et al., 2017) when in fact, as shown here within the Tijuana River Valley, trees of the same species growing under different conditions have different wood characteristics and very different KSHB infestation rates. Therefore, while a pest risk map will be useful in predicting the ultimate range of the KSHB, the Enriched Tree Hypothesis can direct us to sites within the range where the KSHB is likely to have the most impact.

The spread and establishment of the KSHB is likely to be similar to the environmental microbiologist’s tenet, “Everything is everywhere, but the environment selects” (De Wit & Bouvier, 2006). The KSHB may disperse widely but, according to the Enriched Tree Hypothesis, they will establish and thrive only where suitable enriched trees occur. The hypothesis therefore provides the underlying mechanism by which the environment selects. The concept that the KSHB may be everywhere but their establishment at a site depends on the characteristics of the trees at the site is especially important when interpreting the results of shot hole borer trapping efforts because, although the borers may be found in traps and, therefore, present at a site, the trees in the area may not be at high risk of heavy infestation.

Conclusions

The Enriched Tree Hypothesis states that trees of the same species growing under different conditions have different wood characteristics, and the most enriched trees among them are the most susceptible to an abundant KSHB attack. This straightforward hypothesis can explain why the Tijuana River Valley riparian forests were far more severely impacted by the KSHB than other similarly forested sites in the county, and it can predict the general level of impact expected at a site using nutrient enrichment as a major risk factor.

This hypothesis should focus future research on nutrients in the soils and nutrients in the trees because a better understanding of these factors will be essential for predicting the outcome of future shot hole borer attacks in southern California. Finally, the hypothesis can guide management; realizing that nutrient enrichment is a major risk factor, the fertilization and irrigation frequency of urban trees should be reduced where possible to slow tree growth and make trees less attractive to shot hole borers.

Supplemental Information

Supplemental Information 1 Sewage pollution in the Tijuana River and San Diego County.

These data are used in Fig. 2.

Click here for additional data file.

Supplemental Information 2 Wood density data for SAGO and SALA.

These data are used in Figs. 3 and 5, and Table 2.

Click here for additional data file.

Supplemental Information 3 Moisture content data for SAGO and SALA.

These data are used in Fig. 4 and Table 2.

Click here for additional data file.

We thank Elizabeth Onan and Monica Alfaro for field assistance; Monica Almeida at TRNERR for producing the map; and the San Diego Regional Water Quality Control Board, especially Chad Loflen, for the use of its lab. We have benefitted from conversations with Elizabeth Onan, Monica Alfaro and Greg Abbott. This manuscript has been greatly improved by comments from Tiffany Shepherd, Tracy Ellis, Jeff Crooks, Mike Picker, Chad Loflen, Richard Stouthamer, Lawrence Kirkendall, and an anonymous reviewer.

Additional Information and Declarations

Competing Interests

Author Contributions

Field Study Permissions

Data Availability

The authors declare that they have no competing interests. John M. Boland is employed by Boland Ecological Services. Deborah L. Woodward is employed by the California Water Quality Control Board, San Diego Region.

John M. Boland conceived and designed the experiments, performed the experiments, analyzed the data, contributed reagents/materials/analysis tools, prepared figures and/or tables, authored or reviewed drafts of the paper, approved the final draft, field surveys, lab work, project admin.

Deborah L. Woodward conceived and designed the experiments, performed the experiments, contributed reagents/materials/analysis tools, prepared figures and/or tables, authored or reviewed drafts of the paper, approved the final draft, field surveys, lab work.

The following information was supplied relating to field study approvals (i.e., approving body and any reference numbers):

Access permits were received from: San Diego County; US Fish & Wildlife Service; California State Parks; and City of San Diego.

The following information was supplied regarding data availability:

The raw data are available in a Supplemental File.

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
