# Peer review of "Impacts of the invasive shot hole borer (Euwallacea kuroshio) are linked to sewage pollution in southern California: the Enriched Tree Hypothesis"

_PeerJ, doi:10.7717/peerj.6812_

## Round 0.1 · original submission · Major Revisions

The reviewers have made some very good comments that should help improve the manuscript. They did have concerns about the experimental design (perhaps some pseudoreplication) and the rigor of data collection and analysis (perhaps the need to survey "intermediate" sites given that all but 1 of the non-polluted sites is uninfested). Please address these and the other thoughtful comments of the reviewers in your revision.

Reviewer 1 ·

Basic reporting

The manuscript ‘How nutrient enrichment is linked to impacts of the invasive shot hole borer (Euwallacea sp.): the Soft Tree Hypothesis’, presents a hypothesis linking nutrient enrichment (as a consequence of sewage spills) with reduced wood density and increased moisture content of two Salix spp. The hypothesis suggests the resulting ‘soft’ trees are more susceptible to attack by the invasive Kuroshio Shot Hole Borer (KSHB), providing an insight into why infestation and mortality rates have been particularly high in riparian sites in the Tijuana River Valley (TRV).
The manuscript is generally well written and well referenced.
The raw data has been made available, figures are of sufficient resolution however I have some comments:
Figure 1: A and B are missing on the figure- while the figure provides a visual representation of the hypothesis, I think the hypothesis itself is quite clear and do not see that the figure is necessary or adds value to the manuscript.
Figure 3: As with Figure 1, I don’t see that this figure contributes to the manuscript.
Figure 4: A and B missing from the figure
Figures 5 & 6: figure captions should read ‘…mean ± 1 std. dev.’

Experimental design

The manuscript presents original research within the aims and scope of the journal.
The research question is well described, relevant and meaningful, however I have some concerns regarding the rigor of investigation and conclusions drawn (addressed in section 3).

Methods lack details of the statistical package used for analysis.

Validity of the findings

Whilst I believe the hypothesis presented is valid (and I think quite a logical explanation of a situation apparently unique to the TRV), and trust the significance of results relating to wood density and wood moisture content, I have a number of concerns regarding the data supporting significant relationships between level of sewage pollution and infestation and mortality rates.

1. Test # 1 compares sites across a relatively broad spatial scale, with the nine polluted sites in relatively close proximity to each other, and the additional nine non-polluted sites (outside the TR flows) widely dispersed (with the exception of OUT-1, TR tributary). My main concern is that with the exception of site OUT-4 (with 9% infestation), the remaining eight non-polluted sites are entirely without infestation. Whilst this may seem a circular argument (non-polluted sites produce ‘hard’ trees which are not prone to attack), I feel uncomfortably accepting the conclusions drawn under these conditions. It is unfortunate that it was not possible to have a greater number of ‘intermediate’ sites. My concern is that the examined ‘OUT’ sites may have been ‘missed’ by KSHB, and that have 8 of the 9 sites without any sign of attack weakens the argument that infestation rates are driven by pollution. Although I’m not sure how this matter should be resolved.

2. While determining infestation and mortality rates from ‘as many willows as could be examined in two hours’ provides a consistent method by which to score KSHB attack across sites, it results in a high variation in the number of trees assessed (ranging from 34 to 187). No additional information has collected on stand structural traits (such as stem density, basal area and species composition) which may also have a bearing on how easily the sites may be invaded.

3. Adding a ‘severity of infestation’ ranking at the tree level would have allowed a deeper interrogation of wood density/wood moisture content correlation with beetle attack (at the tree level rather than site level).

4. Test # 2, comparison of sites within the TRV, test the Soft Tree Hypothesis by using correlation analysis to independently compare wood density, wood moisture content, KSHB infestation rate and KSHB mortality rate with respect to distance from the main TR channel (as proxy for the amount of sewage reaching each site). I calculated the correlation coefficients for KSHB infestation and mortality rates (vs distance from main channel) using the data provided in Tables 1 and 3. While the r and p values I calculated were consistent with the results presented for infestation rates vs distance (Line 238: r = 0.723, p < 0.05), I found the correlation coefficient for mortality rates vs distance varied to that presented at Line 239 as r = 0.696 and p < 0.05, with my calculation giving r = 0.5074 and p = 0.1344. I am not sure what has happened here, but this is a rather crucial point in terms of suggesting the data provides support for the Soft Tree Hypothesis.

Additional comments

Line 77 – The statement ‘Within a few days mated females emerge …’ is a slightly misleading statement as it seems to suggest the life cycle takes only a few days, rather than weeks to months

Line 327-328 – the suggestion is that only trees impacted by nutrient enrichment and excessive irrigation will be at risk of infestation does not take into account other possible environmental factors which may predispose trees to attack. I find the statement limiting, e.g. storm damage would also surely be among a number of disturbance factors that would predispose trees to attack

Since submission of the manuscript, delineation of the species within the E. fornicatus complex has been completed, KSHB and PSHB can now also be referred to by their scientific names E. kuroshio and E. whitfordiodendrus (Gomez et al., 2018)

Gomez, D.F., Skelton, J., Steininger, M.S., Stouthamer, R., Rugman-Jones, P., Sittichaya, W., Rabaglia, R.J., Hulcr, J., 2018. Species Delineation Within the Euwallacea fornicatus (Coleoptera: Curculionidae) Complex Revealed by Morphometric and Phylogenetic Analyses. Insect Systematics and Diversity 2, 2-2.

While overall my feeling is that the hypothesis is sound and indeed a very likely explanation of the high level of KSHB damage in the TRV, and is an important contribution with regards to where to focus further management efforts, I have some reservations around the soundness of the statistical significance of the data linking sewage levels (using distance from channel as proxy) with KSHB infestation and mortality rates.

Further experimental investigation by conducting pathogenicity trials on excised stems of various densities with the fungal symbiont would be relatively quick and straight forward to conduct. This would allow a well replicated and controlled examination of extent of fungal development in soft vs hard stems, adding wight to support the argument of faster growth of the fungal symbiont

·

Basic reporting

I have read the Boland and Woodward paper with great interest. It is known that two factors will increase the growth of the fungal and beetle population in wood, these are water and nutrients see for instance Gadd 1944 for the influence of nutrients and Bolands earlier work published in PeerJ. It also our experience when we try to grow beetles on logs removed from trees that the water condition of the logs is very important in allowing the beetles to reproduce inside the wood. So both factors are important and may very well explain the high level of infestation found in the Tijuana River Valley. However the points outside of the TRV that are used as controls to determine the influence of sewage spill on the effect of the KSHB, are also areas where the beetle infestation is of a younger age than that found in the TRV, and thus may not yet have resulted in KSHB populations that are equally high. The relative importance of water versus nutrients remains to be determined and could be easily studied experimentally. I think the name soft tree hypothesis is misleading since some of the trees that are attacked and at least loose branches following infestation are woods that are generally considered very hard, coast live oak and sycamore. So I would prefer it the authors could drop that name and replace it with something that reflects water and nutrients to explain the infestation severity. From the literature it appears that both phosphorus and nitrogen are high in the sewage coming from mexico, the relationship between these nutrients in the water, and in the xylem would influence the growth rate of the fungi and consequently that of the beetle. I did a little digging in the literature and it should not be too hard to find papers that make these links. So overall it is very likely that the highly nutrient enriched waters of the Tijuana River spur the growth of the beetle population, through the growth of the fungi, but I doubt that how hard the wood is would influence the beetle population growth.

Experimental design

You may want to clarify when exactly you took the wood samples, you state you did that in fall, is this at the time there were no more leaves on the trees, so that in principle no more xylem was pulled up? Otherwise moisture may accumulate in more heavily infested branches because the water cannot be transported to higher up in the tree, so independent of nutrients, would more heavily infested branches have a higher water content, they would also have lower wood content since part of the wood is removed by the beetles, consequently higher moisture and lower density simply determined by infestation level.

Validity of the findings

No comment

Additional comments

It is very likely that the nutrient availability together with ample water availability determined the food for the fungi and consequently how fast the beetles will be reproducing in the polluted areas. I would prefer you find another name for your hypothesis. Some of the aspects you have brought up could be easily studied experimentally either by supplementing trees with nutrients or water, it would strengthen your arguments. The fact that a large die off has not yet happened at the other locations may be a matter of time since these rivers were invaded later. I would like you to add some literature on the relationship between Phosphorus, nitrogen in ground water, and xylem, and the relationship with growth of Fusarium fungi. I will send you a copy of the gadd paper that you may find helpful.
Line 35 remove native and agriculturally (street trees are also affected)
Line 75 trunks should be trunks and branches
Line 75 with a fungus s/b with fungi.
Line 77 Within in few days mated females emerge. You may want to rephrase that, it takes at least 3 weeks under optimal circumstances to go from egg to adult
Line 132 215-230 degrees F report in Celcius

·

Basic reporting

no comment

Experimental design

see comments in my review text

Validity of the findings

see comments in my review text

Additional comments

PeerJ review of 32720

REVIEWER STRENGTHS AND WEAKNESSES
I have no expertise in tree physiology or in statistics. I do have expertise in bark beetle ecology and behavior.

GENERAL REMARKS
I really enjoyed reading this paper. It is well written and structured, and makes a persuasive argument for an original idea, the Soft Tree Hypothesis. As the authors point out, unique to this hypothesis for why some trees are attacked and not others is that it treats trees not as uniform within species but as varying within species, and explains why some trees in a given species are killed but not others.

I suspect there may be a pseudoreplication issue in that polluted sites are all in one river valley and hence not independently polluted; they share many conditions other than level of pollution. It might be countered, though, by the within-valley analysis.

Correlation is not causation, as we all know. It is important to think critically of what else other than wood density might differ between polluted and nonpolluted habitats, or along a pollution gradient (within-valley analysis). These might or might not be realistic alternative explanations:
• Polluted = flooded (at some time), flooding damages roots and leads to oxygen stress?
• Polluted = flooded, besides nutrients there could be substances (heavy metals? Salts?) which harm trees and weaken their defenses.
• Increased nutrients (in polluted soils) leads not only to more rapid growth and hence less dense wood but maybe (?) increases N content of the wood, which would help ambrosia fungi get established and grow quickly and thus overwhelm trees which are trying to defend themselves.

You should say somewhere how you identified this species, given that it is very difficult.

DETAILED COMMENTS

Re significant digits in reporting results. Too many significant figures given the level of precision, I bet. Also, report SD as one more sig fig than mean: 0.40 ± 0.043. Also, sample size N, correlation coefficient r, and significance level p should be italicized, N, r, p. Standard deviation can be abbreviated SD or S.D., and it only needs to be stated once that the plus-minus refers to S.D. (you repeat this for the second willow species).
98 Study sites. The Tijuana River Valley, San Diego County, California (32o 33.080'N, 117o 4.971'W) is a coastal floodplain of approximately 15 square kilometers at the end of a 4,480 square kilometer watershed.
The precision of the coordinates and indeed the coordinates themselves don’t make sense. The coordinates give one small location of the 15 km2 floodplain.

186 Might be more effective as first sentence rather than last.

297 Importance of N levels treated also in Kirkendall 1983 and Ayres et al. 2000 (refs given below)

REFERENCES
418 Chave, J. 2005 : the link supplied did not work, but I found the pdf at http://www.rainfor.org/upload/ManualsEnglish/wood_density_english[1].pdf
425 Fusarium -> Fusarium
432 “Eskalen A. 2018. Shot hole borers/fusarium dieback websites”: fix Fusarium

FIGURES
Figure 2, map. The blue circles are very difficult to see against the dark green background. Use another color, but not red (avoid red on green, for readers that are red-green colorblind).


Ayres, M. P., Wilkens, R. T., Ruel, J. J., Lombardero, M. J. & Vallery, E. (2000) Nitrogen budgets of phloem-feeding bark beetles with and without symbiotic fungi. Ecology, 81, 2198-2210.
Kirkendall, L. R. (1983) The evolution of mating systems in bark and ambrosia beetles (Coleoptera, Scolytidae and Platypodidae). Zoological Journal of the Linnean Society, 77, 293-352.

---

## Round 0.2 · accepted · Accept

Thank you for your efforts in revising your manuscript in response to the reviewer comments.

# Reviewer 1 ·

Basic reporting

The revised manuscript 'Impacts of the invasive shot hole borer (Euwallacea kuroshio) are linked to sewage pollution in southern California: the Enriched Tree Hypothesis' reads well and has addressed the concerns I had, it provides a valuable contribution to this field of research

Experimental design

I am satisfied that the problem of pseudoreplication has been resolved. The inclusion of intermediate sites (while infested with PSHB rather than KSHB) add value, as does explanation of the statistical methods

Validity of the findings

I am satisfied that the authors have adequately addressed my concerns regarding validity of the findings